# Use of Essential Oils for the Control of Anthracnose Disease Caused by *Colletotrichum acutatum* on Post-Harvest Mangoes of Cat Hoa Loc Variety

**DOI:** 10.3390/membranes11090719

**Published:** 2021-09-20

**Authors:** Luu Thai Danh, Bui Thi Giao, Chau Trung Duong, Nguyen Thi Thu Nga, Doan Thi Kieu Tien, Nguyen Trong Tuan, Bui Thi Cam Huong, Tran Chi Nhan, Dai Thi Xuan Trang

**Affiliations:** 1College of Agriculutre, Can Tho University, Can Tho 94000, Vietnam; btgiao2512@gmail.com (B.T.G.); nttnga@ctu.edu.vn (N.T.T.N.); dtktien@ctu.edu.vn (D.T.K.T.); btchuong@ctu.edu.vn (B.T.C.H.); tcnhan@ctu.edu.vn (T.C.N.); 2Can Tho Technical Economic College, Can Tho 94000, Vietnam; ctduong@ctec.edu.vn; 3College of Natural Sciences, Can Tho University, Can Tho 94000, Vietnam; trongtuan@ctu.edu.vn (N.T.T.); dtxtrang@ctu.edu.vn (D.T.X.T.)

**Keywords:** anthracnose, anti-fungal activity, basil, Cat Hoa Loc, *Colletotrichum acutatum*, cinnnamon, essential oils, lemongrass

## Abstract

Anthracnose disease caused by *Colletotrichum* spp. makes heavy losses for post-harvest mangoes of Cat Hoa Loc variety during storage, packaging, and transportation. The synthetic fungicides are commonly used to control the disease, but they are not safe for consumers’ health and environment. This study was aimed to investigate the use of essential oils (EOs) as the safe alternative control. Pathogen was isolated from the infected Cat Hoa Loc mangoes and identified by morphology and DNA sequencing of the ITS region. Six EOs (cinnamon, basil, lemongrass, peppermint, coriander, and orange) were chemically analyzed by GC–MS. The antifungal activity of EOs was studied in vitro and in vivo. The results showed that the isolated pathogen was *Colletotrichum acutatum*. Cinnamon, basil, and lemongrass EOs effectively inhibited the growth of *C. acutatum* in descending order of cinnamon, basil, and lemongrass. However, they (except basil oil) severely damaged fruit peels. The antifungal activity was closely related to the main compounds of EOs. Basil EOs effectively controlled anthracnose development on Cat Hoa Loc mangoes artificially infected with *C. acutatum*, and its effectiveness was comparable to that of fungicide treatment. Consequently, basil EOs can be used as a biocide to control anthracnose on post-harvest Cat Hoa Loc mangoes.

## 1. Introduction

Vietnam is one of the top mango producers in the world. In 2020, Vietnam’s mango production was around 893,200 tons from 87,000 ha. The main mango cultivation region is in the Mekong Delta, where Cat Hoa Loc variety is considered to be the best, owing to its attractive color, deliciously aromatic taste, and high nutritional value. The domestic and foreign demand for this mango has increased in recent years, resulting in its highest price on the market as compared to other mango varieties grown in Vietnam. However, Cat Hoa Loc mangoes are highly perishable and susceptible to anthracnose disease caused by *Colletotrichum* spp., during storage, packaging, transportation, and marketing [1]. A recent report showed that over 30% of mango fruits in Vietnam were infected with anthracnose [1]. Consequently, any effective control that can delay the symptoms of anthracnose infection would be required to prolong the mango shelf-life.

The control of anthracnose on mangoes mainly relies on the use of fungicides. However, there is increased regulatory restriction of using synthetic fungicides throughout the world, due to concerns about the environmental pollution and adverse effect on consumers’ health. Fungicides, such as prochloraz and guazatine, are commonly applied by passing freshly harvested mangoes through large volume dip solutions. The irresponsible disposal of this hazardous waste can cause serious environmental problems [2]. Fungicide residues left over on fresh fruits gradually accumulated in the human body can trigger a series of health problems for consumers. Furthermore, the frequent use of fungicides would induce pathogen resistance to the active ingredients of fungicides, hence making these synthetic chemicals less effective in controlling diseases. Therefore, the finding of new antifungal agents, that are safe or less toxic to the environment and human health, effectively control pathogens, and have a low risk of pathogens developing resistance, would be extremely valuable. 

One of the feasible and effective approaches is to use essential oils (EOs) as the anti-fungal agents. EOs are mixtures of hundreds of natural volatile compounds derived from plants, mainly monoterpenes, sesquiterpenes, and their oxygenated derivatives [3,4]. Many EOs were demonstrated to possess strong anti-fungal activity [5]. The antifungal activity of EOs may be primarily related to their main components as well as the interaction effect of major and minor components, and hence there is low possibility for pathogens to develop resistance to a wide range of chemicals in EOs. Many EOs are generally recognized as safe (GRAS) and biodegradable. 

Several EOs were shown to effectively control anthracnose caused by various *Colletotrichum* spp. on mango fruits from different varieties: ginger and cinnamon oil on Apple mango variety [6]; lemongrass [7]; basil oil on Willard variety [8]; peppermint oil on Tommy Atkins variety [9]; orange, lemon, and mustard oil on Zabdia variety [10]; and thyme, clove, and cinnamon oil on Banganapalli and Totapuri varieties [11]. Most of these studies did not analyze the chemical composition of EOs, and hence it was quite difficult to convincedly interpret the results. To select EOs that effectively control anthracnose on mango fruits from a specific variety, it is necessary to isolate and identify the causal agent from fruits of this variety and use this isolate for EO testing. Some EOs, even those used at low concentrations, can cause damage to fruit peels, leading to a reduction in market value of the fruits. Consequently, the selection of EOs that are effective in controlling pathogens and have less effect on fruit appearance would be extremely important.

To date, there is no study reported in the literature regarding the use of EOs to control anthracnose disease on mango fruits of Cat Hoa Loc variety caused by *Colletotrichum* sp. Therefore, the objectives of this study were (I) to analyze the chemical composition of six EOs (cinnamon bark, coriander seeds, basil leaves, peppermint leaves, lemongrass stems, and orange peels) by using gas chromatography–mass spectrometry (GC–MS); (II) to isolate and identify the anthracnose-causing agent that is the most pathogenic on Cat Hoa Loc mangoes; (III) to select EOs that have strong antifungal activity in vitro against the isolated fungus; (IV) to determine the effect of selected EOs on Cat Hoa Loc mango fruit peels; and (V) to determine the effect of EOs with negligible impact on Cat Hoa Loc mango peel on controlling anthracnose disease on Cat Hoa Loc mangoes artificially infected by the isolated fungus. 

## 2. Materials and Methods

### 2.1. Sample Collection

The naturally infected mango fruits of Cat Hoa Loc variety were collected from orchards in Hau Giang province, Vietnam. The fruits were kept in the separate sterilized polythene bags and taken to a laboratory for the characteristic examination of anthracnose disease, which was followed by the isolation of the causal agents.

### 2.2. Isolation of Anthracnose-Causing Fungi and Pathogenicity Test

The isolation of the causal fungi was performed for every single fruit. The process of isolating fungi and pathogenicity was performed according to the Koch method described by Agrios [12]. 

The surface of the infected fruit tissues was disinfected with 70% ethanol for 1 min and washed twice with sterilized water. The diseased tissues and surrounding healthy tissues were cut into 5 × 5 mm specimens that were then transferred on water agar (20 g agar and distilled water to 1000 mL) plates, and incubated for 3–7 days at a temperature of 25 °C. The fungal colonies growing from the diseased specimens were inoculated on potato dextrose agar (PDA) plates and incubated for 5 to 7 days. Pure isolates were then obtained by single spore isolation and kept in liquid PDA media at 4 °C for further study. 

Pathogenicity test of the isolated fungi was investigated on the healthy mango fruits of Cat Hoa Loc variety. The fruits were surface sterilized with 70% ethanol for 1 min and wounded by a needle bundle. A total of 100 µL of the fungal spore suspension at a density of 10^8^ colonies/mL was dropped on the wound. The inoculated fruits were stored in nylon bags that were moisturized by cotton soaked in sterile distilled water, and kept at 25 °C. The fruits were examined daily, and the development of anthracnose was recorded. The symptoms present in the artificially infected fruits must be the same as initially observed on the naturally infected fruits. The fungi were then re-isolated, with their morphology and culture needing to be the same as those originally isolated.

### 2.3. Selection of the Most Pathogenic Fungus

The isolated fungi were compared to select the fungal strain causing the most damage to mango fruits of Cat Hoa Loc variety that would be used in the subsequent study. The fresh disease-free fruits with uniform size and shape were artificially infected with various isolated fungi, as described in the above pathogenicity test. Each fungal isolate had 4 replicates with 1 fruit per replicate. The fruits inoculated with sterile distilled water were the control. Diameter of the disease lesions was recorded at 7, 9, 12, and 15 days after inoculation. 

### 2.4. Identification of the Most Pathogenic Fungus

The most pathogenic fungus selected from the previous study was subjected to the identification that was based on the morphological observation and the molecular biological technique. The morphological characteristics of the fungus were compared with the taxonomic study of Sutton [13] to identify the pathogenic fungi. DNA of the fungal sample was extracted in CTAB buffer, and PCR reaction was performed by using universal primer *ITS1* (TCCGTAGGTGAACCTGCGG) and *ITS4* (TCCTCCGCTTATTGATATGC) to amplify the DNA sequence of internal transcribed spacer 1 (ITS1), 5.8S rRNA gene, and ITS2. PCR product was sequenced and compared with sequences published in databases using the BLAST search tool at NCBI to identify the fungal species (http://www.ncbi.nlm.nih.gov/BLAST, accessed on 15 June 2021).

### 2.5. Essential Oil Extraction and Collection

EOs of cinnamon bark, orange fruit peels, lemongrass stems, and basil leaves were extracted by hydro-distillation using a Clevenger-type extraction apparatus [14]. Peppermint leaf and coriander seed EOs extracted by hydro-distillation were purchased from Dalosa Vietnam Essential oils —Herbal Co. Ltd. (Ho Chi Minh, Vietnam). 

### 2.6. Gas Chromatography–Mass Spectrometry Analysis

The chemical components of six EOs were determined according to the method described in the study of Tuan et al. [15] by comparing their Kovats retention index (KI) and mass spectra with those reported in the literature [3,16] and NIST Mass Spectral library. The KI of the components were determined relative to the retention times of a series of n-alkanes (C7–C30). The relative concentrations of these components were obtained from electronic integration measurement on the basis of area normalization of total ion counts. 

The mass spectra of EOs were det ermined by gas chromatography–mass spectrophotometer using a Thermo Scientific (Waltham, MA, USA) system equipped with TB-SQC column (15 m × 0.25 mm × 0.25 μm). Helium was used as a carrier gas with a flux of 1.0 mL/min. The injector and detector temperature were set at 230 and 275 °C, respectively. The oven temperature was programmed at 55 °C for 5 min and increased to 180 °C (at the rate of 3 °C/min), then being increased to 250 °C (at the rate of 10 °C/minutes), which was held for a further 5 min. Sample volume of 1 μL was injected using the split mode (split ratio 1:10).

### 2.7. In Vitro Antifungal Activity of EOs

The antifungal ability of EOs against the isolated and identified *Colletotrichum* sp. was determined by the standard disc diffusion assay [5]. Briefly, EOs were diluted in dimethyl sulfoxide (DMSO) to the concentration of 20 µL/mL. A total of 50 µL of the fungal spore suspension (10^6^ cells/mL) was evenly spread on a Petri dish containing PDA medium, and then the dish was dried. A total of 10 µL of the diluted EO solution was added to the sterile filter paper roll with a diameter of 6 mm, previously placed in a Petri dish containing fungi. The dishes added with 10 µL of 625 µg/mL Talent 50WP (a commercial fungicide with the active ingredient of prochloraz) and 10 µL DMSO were treated as the positive and negative controls, respectively. The dishes were incubated at 30 °C for about 30 min, then incubated at 37 °C for 96 h. The experiment was repeated 5 times for each EO. The zone of inhibition was measured in millimeters after 24, 48, 72, and 96 h of incubation according to the following equation:

Zone of inhibition = Diameter of sterile ring − diameter of filter paper roll.

### 2.8. Determination of Minimum Inhibitory Concentration (MIC) of EOs

The lowest concentrations without visible growth are defined as the minimum inhibitory concentrations (MIC) that completely inhibit the fungal growth. MIC of EOs against *Colletotrichum* sp. was determined by a microdilution technique using 96-well microtiter plates [17]. *Colletotrichum* sp. was cultured on PDA agar plate at 30 °C until spore formation. The fungal spores were washed from the surface of agar plate with 10–20 mL of sterile distilled water. A certain volume of the spore suspension was added on a Neubauer red cell counter (Blaubrand, Germany) that was observed under a microscope for counting the number of spores. Then the volume of spore suspension was adjusted with sterile distilled water to obtain a concentration of approximately 10^6^ spores/mL. A total of 75 µL of the diluted spore suspension was added in wells that contained 75 µL of PDA solution with various EO concentrations and 5% DMSO. There were three control treatments: (1) negative control: 150 µL of liquid PDA medium, (2) positive control 1:75 µL of liquid PDA medium + 75 µL of the diluted spore suspension, (3) positive control 2:75 µL of liquid PDA medium with 5% DMSO + 75 µL of the diluted spore suspension. Each treatment was repeated five times. The 96-well microtiter plates were wrapped with parafilm and incubated at room temperature. The fungal growth in the wells were visually observed by eyes and determined by the optical density measurement at a wavelength of 600 nm using a spectrophotometer (Thermo Scientific, Waltham, MA, USA) at 0 and 24 h after incubation. Three EOs with the lowest MIC would be selected for the subsequent experiment.

### 2.9. Evaluation of the Effect of EOs on Mango Fruit Peels 

The physiological mature mango fruits of Cat Hoa Loc variety were purchased from growers in Hau Giang province, Vietnam. The healthy fruits that were uniform in size and peel color were dipped in solutions containing EOs at a concentration equal to MIC pre-determined from the previous experiment for 30 s, and left at 30 °C for 4 h. The visual observation was performed to select EOs that had the least impact on the mango fruit peels, for the next experiment. There were three replicates for each treatment and one mango for each replicate.

### 2.10. Effect of EOs on Inhibiting Anthracnose Development on Mango Fruits

The inhibitory ability of EOs against the development of anthracnose on mango fruits of Cat Hoa Loc variety was determined as described in the study of Le et al. [18], with some modifications. The disease-free fruits with uniform ripeness and size were washed with tap water, then sterilized with 70% ethanol, and allowed to air-dry. The fruits were wounded at four positions with a depth of 2 mm by a bundle of needles. A total of 15 µL of EOs in 5% DMSO solution with the concentration equal to MIC pre-determined in the previous experiment was applied on the wounds and left to dry for 1 h. Subsequently, 15 µL of spore suspension (10^6^ spore/mL) was added on the treated wounds and left to dry for 2 h. The fruits treated with distilled water, DMSO, and Talent 50WP (Prochloraz) were the control treatments. Each treatment had 10 replicates with 1 fruit per replicate. The fruits were stored at room temperature for 7 days. Diameter of disease lesions was measured at 3, 4, 5, 6, and 7 days after inoculation. 

### 2.11. Statistical Analysis

Data were processed by using IBM SPSS Statistics for Windows, version 21 (IBM Corp., Armonk, NY, USA) for one-way analysis of variance. The means of treatments were compared by using Duncan’s test at 5% level of significance.

## 3. Results and Discussion

### 3.1. Chemical Components of EOs

Chemical components and their percentages of the total EO content in six EOs determined by GC–MS are presented in Table 1 and Table 2. Nine chemical compounds were identified from basil EOs with the total percentage of 95.1%, in which methyl chavicol (61.3%) and linalool (24.8%) were the main components (Table 1). This finding was in agreement with the study of Oxemham et al. [19], Grayer et al. [20], Marotti et al. [21], and Sishu et al. [22]. Basil varieties could be classified into two groups on the basis of EO chemotype: methyl chavicol and linalool [19]. Methyl chavicol and linalool chemotye contain methyl chavicol and linalool, respectively, as the most abundant component in EOs [19,20,21]. Basil EOs in this study could be considered as methyl chavicol chemotype. In the study of Oxemham et al. [19], the methyl chavicol chemotype contained methyl chavicol (76.1%) and linalol (18.6%), while linalol chemotype had linalol, eugenol, eucalyptol, and caryophyllene of 53%, 12.4%, 7.7%, and 5% of the whole oil, respectively.

A total of 13 chemical components accounting for 89.1% of cinnamon bark EOs were determined (Table 1). Trans-cinnamaldehyde (43.3%), safrole (26.7%), cis-cinnamaldehyde (10.9%), and 1,8-cineole (4.5%) were the main constituents of EOs. Commercial cinnamon EOs are commonly extracted from the bark of three cultivated species, namely, *Cinnamomum cassia*, *C. loureirii*, and *C. verum* synonym *C. zeylanicum* [23]. Bark EOs of *C. cassia* have trans-cinnamaldehyde (42.4–90.6%), cinnamic acid (5.5–43.1%), methoxy-cinnamaldehyde (3.5–13.8%), and cynnamyl acetate (1.0–5.4%) as the main compositions [24,25,26,27,28,29,30]. *C. loureirii* bark EOs commonly contains four main components, namely, trans-cinnamaldehyde (50.2–92.9%), α-copaene (2.1–21.3%), α-guaiene (3.4–9.3%), and β-cadinene (4.1–7.7%) [31,32,33,34,35,36]. *C. verum* varieties are grouped into two chemotypes according to their main components in EOs: eugenol and safrole chemotype [23,36]. Bark EOs of eugenol chemotype usually have four marker compounds, namely, trans-cinnaldehyde (50.5–89.3%), cinnamyl acetate (1.5–8.8%), eugenol (0.4–8.8%), and 1,8-cineole (1.0–4.6%) [24,32,34,37,38]. Safrole chemotype often contains two fingerprint compounds in bark EOs, namely, trans-cinnamaldehyde (52.5–74%) and safrole (2–10.8%) [23,39]. In addition, other species from *Cinnomomum* genus (*C. pubescens*, *C. impressicostatum*, *C. mollissimum*, *C. porrectum*, and *C. camphora*) have a high content of safrole (3.2–93.4%) in their bark EOs; however, they have no content of trans-cinnamaldehyde [40]. From the literature review of EOs extracted from barks of cinnamon species, it can be suggested that cinnamon EOs in this study with two main marker compounds identified (trans-cinnamaldehyde and safrole) belong to the safrole chemotype of *C. verum.*


There were 16 chemical compounds determined in lemongrass EOs, accounting for 87.3% of the total oil content (Table 1). Geranial (34.6%), neral (34.5%), geraniol (3.4%), geranyl acetate (2.6%), and cis-carveol (2.0%) were the main components of lemongrass EOs. The results were in agreement with the chemical composition of *Cymbopogon citratus* EOs reported in several studies [3,41,42,43,44]. A natural mixture of two isomeric acyclic monoterpene aldehydes, geranial (transcitral, citral A) and neral (cis-citral, citral B), is named as citral, being the main chemical component of lemongrass oil. Geranial and neral have the same molecular formula, C_10_H_16_O, but have differently chemical structures. 

EOs extracted from the fruit peels of “Cam Sanh” orange had eight chemical compounds identified with a total of 88.7% (Table 2). Limonene (87.2%) and β-myrcene (0.9%) were the main components of “Cam Sanh” orange peel oil. Cam Sanh orange is the hybrid species of *Citrus reticulata* and *Citrus sinensis,* with the scientific name of *Citrus reticulata x sinensis*. There is very limited information reported in the literature about the chemical composition of EOs extracted from this hybrid. However, EOs from fruit peels of *C. reticulata* and *C. sinensis* have been extensively documented. *C. reticulata* oils have four main chemical components, namely, limonene (65.3–74.2%), γ-terpinene (16.4–22.7%), α-pinene (2.0–2.7%), and β-pinene (1.4–2.1%), while *C. sinensis* oils have limonene (83.9–95.9%) and β-myrcene (0.9–3.3%) as the main components [3]. It can be seen that the chemical composition of Cam Sanh orange peel EOs is closely related to that of *C. sinensis.*

A total of 12 chemical compounds identified from peppermint (*Mentha piperita*) EOs accounted for 92.0% of oil content (Table 2). Carvone (61.6%), pulegone (12.3%), and limonene (6.2%) were the main compositions of peppermint oil. Peppermint EOs can be sorted into two chemotypes according to their content of carvone and menthol. Carvone chemotype contains three main compounds, namely, carvone (34.9–84.3%), pulegone (0.8–14.8%), and limonene (8.1–11.2%), while menthol chemotype has three different dominant compounds, namely, menthol (22.3–54.2%), menthone (9.1–39.3%), and menthyl acetate (2.0–15.1%) [45,46,47,48,49,50]. Consequently, peppermint oil in this study may belong to carvone chemotype. 

Coriander (*Coriandrum sativum*) fruit EOs had 21 chemical compounds identified, with a total of 74.9% oil content (Table 2). Linalool (55.3%), neryl acetate (4.3%), γ-terpinene (3.1%), and p-cymene (2.6%) were the main constituents of coriander fruit oil. The content of these compounds was within the content range reported in the literature [3,51,52,53,54] regarding coriander fruit EOs with linalool (36.7–87.5%), neryl acetate (0–8.4%), γ-terpinene (0.1–9.1%), and p-cymene (2.6%). 

The chemical compositions of EOs extracted from the same part of the same plant species are not the same across the different studies reported in the literature. This could be due to the fact that the chemical constituents of plant EOs are greatly affected by five main factors: genetics (varieties), environment (geographical, climatic, seasonal, and soil conditions of growing location), cultivation (irrigation, fertilization, harvesting time, stage of maturity), sample preparation (drying methods, fresh or dry materials, particle size of plant samples), and extraction methods (solvent extraction, hydro-distillation, CO_2_ extraction, microwave-assisted extraction).

### 3.2. Isolation of the Most Pathogenic Fungus

The isolation of the fungal strains causing anthracnose disease on the fruits of Cat Hoa Loc variety was based on the disease lesions described in the study of Le Hoang Le Thuy and Pham Van Kim [55]. During the process of collecting the infected fruits, the disease symptoms were mainly observed on young leaves, flowers, and fruits. The initial symptoms were from small brown to black spots. There were many disease spots on the leaves, and these spots joined together to form large, brown to black lesions, surrounded by dark brown margins. Young leaves could not develop and consequently affect plant growth. The infected flowers were black and could not bear fruit. The diseased fruits had irregular black spots on the peel that caused fruit drop or affected post-harvest fruit quality. From over 10 fungal strains isolated from the infected fruits, four strains (namely, Col 1, Col 2, Col 5, and Col 6) were artificially demonstrated to cause anthracnose on mango fruits, showing dark brown and black lesions.

Four pathogenic strains were compared to select the strain that caused the most damage to mango fruits. The diameter of disease lesions on Cat Hoa Loc mango fruits caused by four fungal strains that was measured at 6, 9, 12, and 15 days after artificial infection (D.A.I.) is shown in Table 3. Initially, most of the lesions were round, while a few were irregular shape. Subsequently, they were dry with dark brown color and slightly concave, having no discernible border (Figure 1). At 6 D.A.I., all fungal strains showed pathogenicity with lesion diameter ranging from 1.8 to 9.9 mm, in which Col 5 strain induced the largest diameter of 9.9 mm. The damage caused by four fungal strains on mango fruits increased over time. At 15 D.A.I., the strains of Col 1, Col 5, and Col 6 showed the highest diameters of 19.2, 20.0, and 16.2 mm, respectively, with there being no statistically significant difference. It can be seen that the Col 5 strain caused the most damage to mango fruits as compared to the remaining strains (Table 3, Figure 1). Therefore, it was selected for the identification and use in the following experiments.

### 3.3. Identification of the Most Pathogenic Fungus

The cultural and morphological characteristics of the studied fungus are presented in Figure 2. The colony produced the aerial mycelium with a smooth filamentous form and even edge. The mycelial growth was relatively slow—after 9 days of inoculation, the mycelial diameter in the Petri dish was about 5.5 cm. Mycelium of *C. acutatum* reached the edge of the Petri dish at about 14 days, while the mycelial growth rate of *C. gloeosporioides* was relatively fast and reached the edge of the Petri dish at about 10 days [56]. The upper colony surface was white, while the lower colony surface had a light orange color. The fungal conidia were fusiform and cylindrical with an acute end. Conidiophores were mostly simple, sometimes branched. The characteristics of the studied isolate causing anthracnose on mango in this study were consistent with the description of Sutton [13] regarding *Colletotrichum* spp. Furthermore, these characteristics were similar to those of *Colletotrichum acutatum* described in the studies of Khan et al. [56], Jayasinghe et al. [57], Sundelin et al. [58], and Peres et al. [59].

The rDNA sequence of ITS1 and ITS2 regions including the 5.8S ribosomal subunit of the isolate was successfully amplified with a length of 586 nucleotides by using the universal primer pair of *ITS1* and *ITS4*. The similar length of this region was also reported in the study of Shi et al. [60], who amplified DNA extracted from *Colletotrichum acutatum* by using *ITS1* and *ITS4* primers. BLAST similarity search showed that the sequence of ITS1, 5.8S rRNA, and ITS2 from the studied isolate shared the sequence identity of 99% with the published ITS sequences of *Colletotrichum acutatum* in Genbank, with the accession number of AJ749675.1. 

From the cultural, morphological, and molecular biological evidence presented above, we can conclude that the studied isolate causing anthracnose disease on Cat Hoa Loc mango fruits is *Colletotrichum acutatum*. This fungus is one of the most frequently reported species of the genus and causes diseases commonly known as anthracnose on numerous host plants worldwide. It has been known to be especially destructive on fruits such as papaya and capsicum, strawberry, citrus, apple, olive, cranberry, blueberry, and mango. This fungal species has been demonstrated to be the most infectious, because it is resistant to a wide range of fungicides, namely, carbendazim [57], benomyl and copper oxide [59], quinone outside inhibitors [61], and mancozeb and thiophanate-methyl [62]. 

### 3.4. In Vitro Antifungal Activity of EOs against C. acutatum

The antifungal activity of six EOs assayed by the agar diffusion method is shown in Table 4 and Figure 3. A total of 10 µL of EO solution in DMSO with a concentration of 20 µL/mL or control (100% DMSO or 625 µg/mL Talent 50 WP) was tested against mycelial growth of *C. acutatum* on an agar plate. The antifungal activity was determined by measuring the diameter of clear inhibitory zones appearing against a white background on the agar plate. These clear zones represented regions where fungal mycelia or reproductive stroma were not present [63]. Among the tested EOs, the mean fungal growth inhibition of cinnamon bark EOs was the largest over 4 D.A.I., followed in descending order of basil, lemongrass, peppermint, and coriander seed oils. Orange peel oil showed no antifungal activity against *C. acutatum*. Similarly, DMSO treatment had no growth inhibition recorded over 4 D.A.I., indicating that DMSO did not contribute to the antifungal activity of EO solution in DMSO. 

The antifungal activity of all tested EOs was greatly reduced over 4 D.A.I., while that of prochloraz was almost unchanged in the same period. Across 4 days of observation, at the first D.A.I., the growth inhibition of all EOs and fungicide was the largest; however, at the second D.A.I., the reduction in inhibition was observed in all treatments (except orange EOs and DMSO treatment). At the third and fourth D.A.I., the antifungal activity of fungicide was unchanged as compared to that at the second DAI; however, all EO treatments showed significant decrease over the same period. Similar results were also found in the study of Morkeliūnė et al. [64] in that the inhibitory activity of sage and peppermint EOs against *C. acutatum* was lower at 7 D.A.I. as compared to that of 4 D.A.I. In the agar diffusion assay, EOs or fungicide solution was added on the filter paper placed onto the agar disc that had the surface completely covered by the fungal spore suspension. EOs and fungicide gradually diffused away from the filter paper. The further distance from the filter paper, the less concentration of EOs and fungicide. At the first D.A.I., not many fungal spores germinated, leading to low mycelial growth; therefore, EOs and fungicide at low concentration was very effective, resulting in the largest growth inhibition recorded in all treatments. However, at the second D.A.I., the mycelia development was vigorous, and EOs and fungicide became ineffective at low concentrations, leading to the reduction in the fungal growth inhibition. Beyond this date, further reduction was observed in all EO treatments, but not in fungicide treatment. It can be explained by the fact that EOs are volatile while prochloraz is non-volatile [65]. Over time, the chemical components of EOs slowly escaped into the air, resulting in lower concentrations of EOs in agar disc, and therefore EOs treatments had less of an inhibitory effect on the mycelial growth. On the other hand, the unchanged concentration of fungicide in agar disc owing to its non-volatile property maintained the effectively inhibitory activity against the fungal development. 

MIC of six EOs against *C. acutatum* is presented in Table 5. The lower the MIC value of EOs, the higher the antifungal activity. MIC of cinnamon bark EOs was the lowest (1.6 µL/mL), followed by basil (4 µL/mL) and lemongrass (12 mL) EOs. MIC of orange, peppermint, and coriander EOs was greater than 32 µL/mL. The antifungal activity of EOs expressed by MIC was consistent with that indicated by the fungal growth inhibition in Table 4. Cinnamon bark EOs had the strongest antifungal activity, followed in descending order by basil, lemongrass, peppermint, coriander, and orange EOs. Similarly, the antifungal activity against *C. acutatum* causing grape ripe rot of cinnamon (*Cinnamomum cassia*) EOs was the strongest as compared to that of holy basil and peppermint EOs [66]. The study of Morkeliūnė et al. [64] also reported that peppermint and coriander EOs had weak antifungal activity against *C. acutatum*, causing anthracnose on strawberry fruits.

The antifungal activity of cinnamon EOs may be primarily related to their major components. The major constituents of cinnamon EOs were found to be trans-cinnamaldehyde (43.3%), safrole (26.7%), cis-cinnamaldehyde (10.9%), and 1,8-cineole (4.5%) (Table 1). Trans-cinnamaldehyde was demonstrated to be a potent antifungal agent [67,68,69]. Safrole was shown to possess a moderate inhibitory activity against *C. acutatum* isolated from the infected tamarillo. 1,8-Cineole had no inhibitory activity against *C. acutatum* isolated from ripe rot diseased grapes [70]. Therefore, the antifungal activity of cinnamon bark oil may be mainly due to trans-cinnamaldehyde. The higher the trans-cinnamaldehyde content in cinnamon oil, the stronger the antifungal activity of cinnamon oil expressed by its lower MIC against *C. acutatum*. It is further supported by comparing MIC and trans-cinnamaldehyde content of cinnamon oil in this study with those reported in the study of Duduk et al. [71] and He et al. [72]. MIC of cinnamon oil in this study was the highest (MIC = 1.6 µL/mL), while its trans-cinnamaldehyde content was the lowest (43.3%). In contrast, cinnamon bark oil had the lowest MIC (0.2 µL/mL) against *C. acutatum*, causing anthracnose on kiwifruit, with the highest cinnamaldehyde content (86.2%) [72]. Cinnamon bark oil had the moderate MIC (0.67 µL/mL) against *C. acutatum* causing anthracnose on strawberry fruits and moderate trans-cinnamaldehyde content (73%) [71].

The strong antifungal activity of basil oil may be due to its main compositions, namely, methyl chavicol (61.3%) and linalool (24.8%) (Table 1). Methyl chavicol was proven to be a strong antifungal agent against *Moniliophthora perniciosa* with minimum fungicidal concentration (MFC) of 1000 ppm [73] and *Uromyces viciae-fabae* with MFC of 1000 ppm [19]. Similarly, linalool showed the strong inhibitory activity against *Uromyces viciae-fabae* with MFC of 1000 ppm [19]. MFC is defined as the concentration of EOs, resulting in the death of 99.9% of the fungi. Methyl chavicol and linalool significantly reduced the mycelial growth of *Botrytis fabae* on solid media at concentrations of 1000 and 600 ppm, respectively, at 4 days after inoculation [19]. However, linalool was demonstrated to be ineffective in controlling the growth of three *Colletotrichum* species, namely, *C. acutatum, C. fructicola*, and *C. gloeosporioides* [67,70]. Consequently, the antifungal activity of basil oil against *C. acutatum* in this study may be primarily from methyl chavicol. 

Lemongrass oil had strong antifungal activity against *C. acutatum* due to its citral content (69.1%) (Table 1). Citral is a strong antifungal agent against many species of plant pathogenic fungi. It completely inhibited the mycelial growth of *Colleotrichum fructicola* and *Colleotrichum acutatum* isolated from ripe rot diseased grapes at concentration of 1 mg/mL [70]. Furthermore, MIC of citral against *Colletotrichum musae*, *Colletotrichum gloeosporioides*, and *Fusarium subglutinans* was very low (0.6%) [74]. Volatile citral completely suppressed the conidial germination of *C. gloeosporioides* isolated from the infected pepper fruit at the concentration of 2 µL/disc [67]. Citral in the vapor form totally controlled the mycelial growth of *Colletotrichum lindemuthianum*, *Fusarium oxysporum*, and *Botrytis cinerea* at a very low concentration of 10 ng/mL in the atmosphere [75].

Peppermint, coriander, and orange EOs had negligible or no antifungal activity against *C. acutatum* (Table 4 and Table 5). It may be due to the fact that their main components have low antifungal activity. Carvone (61.6%), linalool (55.3%), and limonene (87.2%) were the main components of peppermint, coriander, and orange EOs, respectively (Table 2). Carvone was not effective against *Fusarium subglutinans* but had low inhibitory activity against *C. musae* and *C. gloeosporioides* with MICs of 0.8 and 1%, respectively [74]. Linalool and limonene were ineffective in inhibiting the mycelial growth of *C. fructicola* and *C. acutatum* isolated from ripe rot diseased grapes at a concentration of 1 mg/mL [70]. In the vapor form, carvone and limonene had low antifungal activity against the mycelial growth of *C. lindemuthianum*, *F. oxysporum*, and *B. cinerea* [75]. Volatile linalool and limonene could not completely suppress the conidial germination of *C. gloeosporioides* isolated from the infected pepper fruit at the concentration of 8 µL/disc [67]. 

### 3.5. Effect of EOs on Fruit Peel of Cat Hoa Loc Mangoes

Three EOs, namely, cinnamon, basil, and lemongrass oil, were selected to evaluate their effect on mango fruit peels. MIC of EOs was used in this study: cinnamon (1.6 µL/mL), basil (4 µL/mL), and lemongrass (12 µL/mL). Mango fruits of Cat Hoa Loc variety were dipped into EOs solution containing 5% DMSO and left to air-dry. Fruit peels were observed after 4 h of treatment. The results are presented in Figure 4. Lemongrass treatments showed heavily damaged fruit peel because its tested concentration was too high at up to 12 µL/mL. However, cinnamon oil was tested at a very low concentration (1.6 µL/mL), and it still induced an adverse effect on mango fruit peel. The finding may indicate that mango fruits of Cat Hoa Loc variety are sensitive to cinnamon bark. Basil treatment had a negligible effect on fruit peel, even though it was tested at quite a high concentration of 4 µL/mL. Cinnamon and lemongrass oil may be effective in controlling anthracnose disease, but they severely affect sensory value of mango fruits. Therefore, only basil oil was selected for the subsequent in vivo study.

### 3.6. Effect of Basil Oil on Anthracnose Development on Mango Fruits

Anthracnose lesion diameter under effect of basil oil, fungicide, and control treatment over 7 days after inoculation is shown in Table 6 and Figure 5. Lesion diameter in all treatments increased over time. The higher the lesion diameter, the lower the antifungal activity. The inhibitory effect of basil oil treatment was slightly lower than that of fungicide over 7 D.A.I.; however, the effect of both treatments was not significantly different. In addition, basil oil treatment had the suppressive activity against anthracnose development, nearly double that of both control treatments (5% DMSO and distilled water). The findings indicated that basil oil is effective against anthracnose development on mango fruits of Cat Hoa Loc variety artificially inoculated with *C. acutatum*. Similarly, the study of Karunanayake et al. [8] also found that basil oil incorporated in beeswax coating at low concentration of 0.4 µL/mL was very effective in increasing shelf-life and reducing anthracnose development in mango fruits of Willard cultivar. Treatment with basil oil controlled crown rot and anthracnose of banana, with no detrimental effect on organoleptic properties [76]. 

## 4. Conclusions

The study isolated and identified the most pathogenic fungus, *Colletotrichum acutatum*, which causes anthracnose disease on Cat Hoa Loc mangoes. GC–MS analysis determined the main components of cinnamon (trans-cinnamaldehyde 43.3%), basil (methyl chavicol 61.3%), lemongrass (citral 69.1%), peppermint (carvone 61.6%), coriander (linalool 55.3%), and orange (limonene 87.2%) EOs that may contribute to the anti-fungal activity of EOs. Peppermint, coriander, and orange EOs had negligible or no inhibitory effect on *C. acutatum*. Cinnamon (MIC = 1.6 µL/mL), basil (MIC = 4 µL/mL), and lemongrass (MIC = 12 µL/mL) EOs showed strong antifungal activity against *C. acutatum*. As used at MIC, lemongrass and cinnamon EOs caused severe damage on fruit peels of Cat Hoa Loc mangoes, while basil EO had negligible impact. Basil oil effectively controlled anthracnose development on Cat Hoa Loc mangoes artificially infected with *C. acutatum*. The disease control effectiveness of basil EO was comparable to that of fungicide treatment. It can be concluded that basil EO with methyl chavicol as the main composition can be applied as a bio-based agent to protect post-harvest mangoes of Cat Hoa Loc variety from anthracnose disease caused by *C. acutatum* that in turn help to extend mango shelf-life during storage, packaging, transportation, and marketing. 

## Figures and Tables

**Figure 1 membranes-11-00719-f001:**
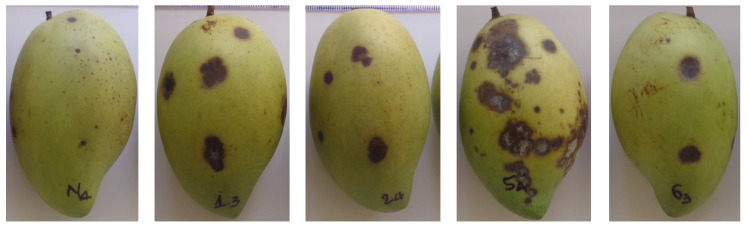
Anthracnose symptoms on Cat Hoa Loc mango fruits artificially infected by sterile distilled water, Col 1, Col 2, Col 5, and Col 6 strains (from **left** to **right**) at 15 days after artificial infection.

**Figure 2 membranes-11-00719-f002:**
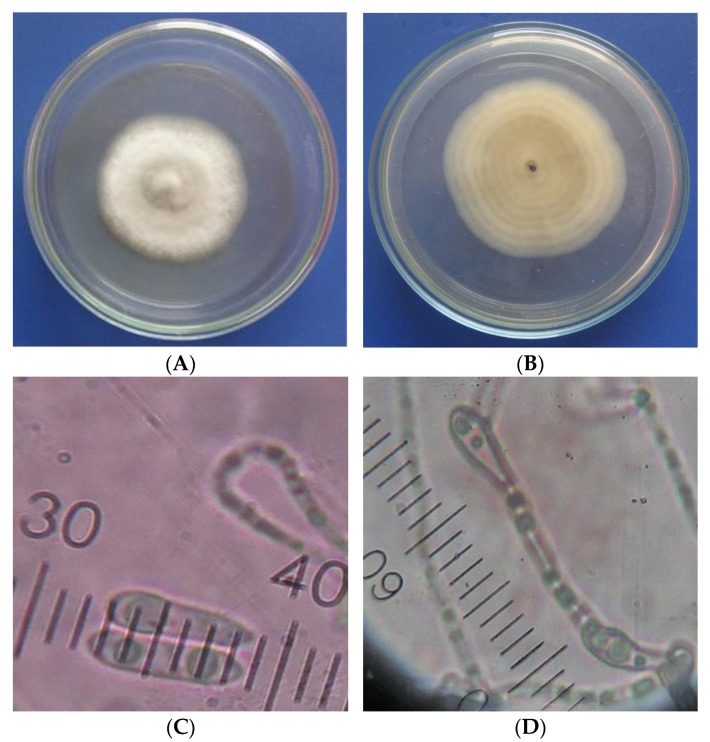
The cultural and morphological characteristics of *Colletotrichum* sp. causing anthracnose on Cat Hoa Loc mango fruits at 9 days after inoculation. (**A**) Upper colony surface; (**B**) lower colony surface; (**C**) conidia; (**D**) conidiophores.

**Figure 3 membranes-11-00719-f003:**
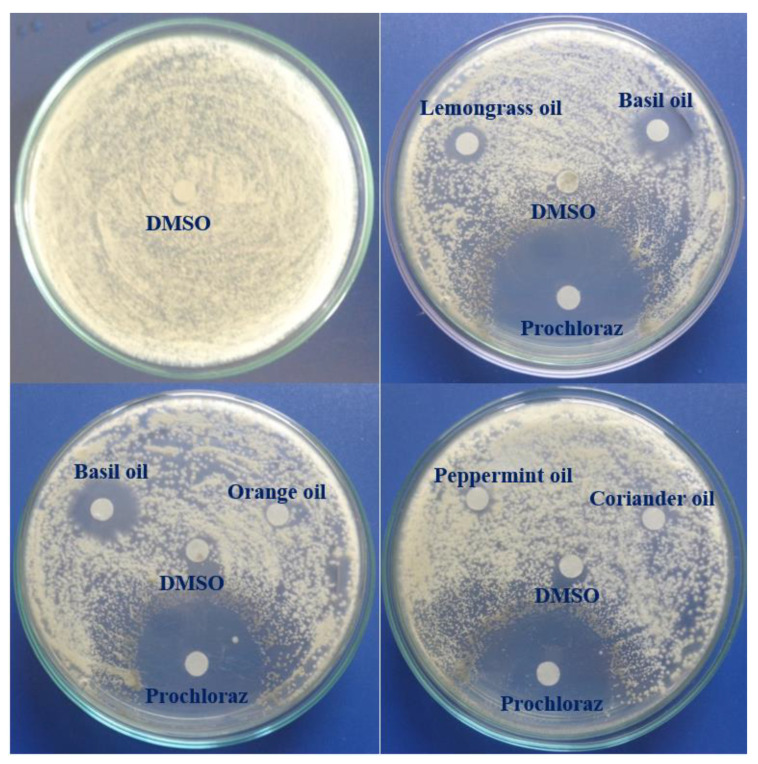
The antifungal activity of EOs, dimethyl sulfoxide (DMSO), and fungicide (prochloraz) against the mycelial growth of *C. acutatum* at 4 days after innoculation.

**Figure 4 membranes-11-00719-f004:**
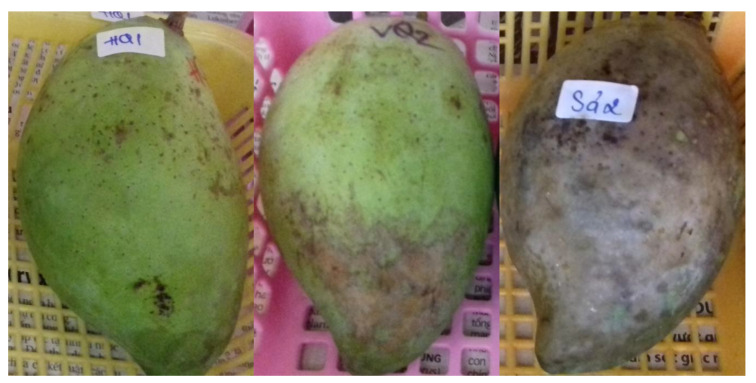
Effect of EOs on the peels of Cat Hoa Loc mango fruits. From left to right: basil, cinnamon, and lemongrass treatment.

**Figure 5 membranes-11-00719-f005:**
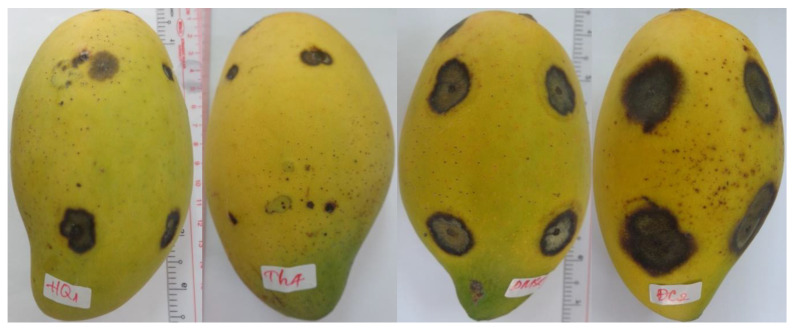
Anthracnose lesion on mango fruits of Cat Hoa Loc variety under effect of different treatments at 7 D.A.I. From left to right: basil EOs, fungicide, DMSO, and distilled water treatment.

**Table 1 membranes-11-00719-t001:** Chemical components of lemongrass, cinnamon bark, and basil essential oils (EOs).

	Lemongrass Oil			Cinnamon Bark Oil			Basil Oil		
No.	KI	Compound	%	KI	Compound	%	KI	Compound	%
1	975	6-Methylhelpt-5-en-2-one	0.84	1023	1,8-Cineole	4.54	1029	1,8-Cineole	1.10
2	984	β-Myrcene	0.32	1113	Linalool	0.24	1102	Linalool	24.80
3	1027	trans-β-Ocimene	0.63	1158	Borneol	0.71	1205	Methyl chavicol	61.31
4	1103	Linalool	1.60	1219	(Z)-Cinnamaldehyde	10.90	1241	Neral	0.67
5	1147	Lavandulol	0.29	1275	(E)-Cinnamaldehyde	43.27	1272	Geranial	0.84
6	1155	Citronellal	0.65	1278	Safrole	26.69	1407	α-Cedrene	3.40
7	1167	trans-Chrysanthenol	0.50	1404	α-Cedrene	0.16	1430	β-Copaene	1.16
8	1186	Isogeraniol	1.79	1416	β-Copaene	0.45	1470	γ-Gurjunene	0.66
9	1205	trans-Carveol	0.31	1454	trans-α-Bisabolene	0.43	1505	β-Bisabolene	1.11
10	1218	cis-Carveol	1.98	1459	δ-Cadinene	0.24	-	-	-
11	1249	Neral	34.51	1663	Tetradecanol	0.72	-	-	-
12	1261	Geraniol	3.37	1698	epi-α-Bisabolol	0.41	-	-	-
13	1283	Geranial	34.62	1831	2-Phenylethyl benzoate	0.34	-	-	-
14	1387	Geranyl acetate	2.63	-	-	-	-	-	-
15	1408	E-Caryophyllene	1.66	-	-	-	-	-	-
16	1571	Caryophyllene oxide	1.56	-	-	-	-	-	-
	**Total % identified**		**87.26**			**89.10**			**95.05**

**Table 2 membranes-11-00719-t002:** Chemical components of coriander fruit, peppermint leaf, and orange fruit peel EOs.

	Coriander Fruit Oil			Peppermint Oil			Orange Fruit Peel Oil		
No.	KI	Compound	%	KI	Compound	%	KI	Compound	%
1	981	β-Pinene	0.63	930	α-Pinene	0.69	910	α-Thujene	0.06
2	1000	n-Octanal	0.14	1030	Limonene	6.22	983	β-Myrcene	0.87
3	1025	p-Cymene	2.56	1145	Menthone	1.72	1033	Limonene	87.23
4	1028	α-Myrcene	2.82	1161	Menthofuran	1.78	1104	n-Nonanal	0.17
5	1058	γ-Terpinene	3.09	1169	Menthol	2.48	1145	Limonene oxide	0.07
6	1073	cis-Linalool oxide	0.11	1233	Pulegone	12.29	1189	α-Terpineol	0.09
7	1087	Terpinolene	0.18	1240	Carvone	61.55	1207	Decanal	0.19
8	1113	Linalool	55.33	1250	Piperitone	3.54	1388	Geranyl acetate	0.03
9	1141	Camphor	0.72	1435	γ-Elemene	0.21	-	-	-
10	1154	Borneol	0.20	1510	Germacrene A	0.11	-	-	-
11	1176	Terpinene-4-ol	0.50	1583	Caryophyllene oxide	0.62	-	-	-
12	1190	α-Terpineol	0.05	1644	α-Muurolol	0.74	-	-	-
13	1231	Nerol	0.18	-	-	-	-	-	-
14	1234	Citronellol	0.15	-	-	-	-	-	-
15	1260	Geraniol	1.37	-	-	-	-	-	-
16	1349	Myrtenyl acetate	0.54						
17	1358	Citronellyl acetate	1.73						
18	1369	Neryl acetate	4.25						
19	1409	β-Caryophyllene	0.11						
20	1468	2-Dodecenal	0.23						
21	1519	γ-Cadinene	0.05						
**Total % identified**			**74.94**			**91.95**			**88.71**

**Table 3 membranes-11-00719-t003:** Diameter of anthracnose lesions caused by four fungal strains over time.

Fungal Strain	Diameter of Disease Lesions (mm) Recorded over Time			
	6 D.A.I.	9 D.A.I.	12 D.A.I.	15 D.A.I.
Col 1	5.10 ^b^	11.7 ^ab^	16.5 ^a^	19.2 ^ab^
Col 2	1.80 ^bc^	8.65 ^b^	11.9 ^b^	13.8 ^b^
Col 5	**9.90 ^a^**	**13.1 ^a^**	**15.5 ^ab^**	**20.0 ^a^**
Col 6	4.70 ^b^	9.55 ^ab^	11.6 ^b^	16.2 ^ab^
Sterile DW	0.00 ^c^	0.00 ^c^	0.00 ^c^	1.70 ^c^

Note: D.A.I., days after artificial infection. DW, distilled water. Numbers followed by the different letters in the same column were significantly different at 5% significance level by using Duncan’s test.

**Table 4 membranes-11-00719-t004:** The antifungal activity of six tested EOs against *C. acutatum*.

Treatment	Mean Fungal Growth Inhibition (mm)			
	1 D.A.I.	2 D.A.I.	3 D.A.I.	4 D.A.I.
Lemongrass	**3.1 ^d^**	**1.8 ^c^**	**1.4 ^c^**	**1.1 ^c^**
Cinnamon bark	**14.2 ^b^**	**5.2 ^b^**	**4.0 ^b^**	**2.8 ^b^**
Basil	**6.0 ^c^**	**4.7 ^b^**	**3.6 ^b^**	**3.2 ^b^**
Orange peel	0.0 ^f^	0.00 ^d^	0.00 ^e^	0.00 ^d^
Peppermint	1.4 ^e^	0.2 ^d^	0.8 ^cd^	0.00 ^d^
Coriander seed	0.9 ^ef^	0.3 ^d^	0.00 ^e^	0.00 ^d^
Prochloraz	**17.0 ^a^**	**15.8 ^a^**	**15.8 ^a^**	**15.8 ^a^**
DMSO	**0.00 ^f^**	**0.00 ^d^**	**0.00 ^e^**	**0.00 ^d^**

Note: D.A.I., days after artificial infection. Numbers followed by the different letters in the same column were significantly different at 5% significance level by using Duncan’s test.

**Table 5 membranes-11-00719-t005:** Minimum inhibitory concentration (MIC) of EOs against *C. acutatum*.

EOs	MIC (µL/mL)
Cinnamon bark	1.6
Basil	4
Lemongrass	12
Orange peel	>32
Peppermint	>32
Coriander	>32

**Table 6 membranes-11-00719-t006:** Effect of basil oil, fungicide, and control treatment on lesion diameter (mm) of anthracnose on mango fruits caused by *C. acutatum* over 7 days after inoculation.

Treatment	Lesion Diameter (mm) over Time				
	3 D.A.I.	4 D.A.I.	5 D.A.I.	6 D.A.I.	7 D.A.I.
Basil oil	**2.25 ^bc^**	**4.40 ^bc^**	**5.62 ^b^**	**8.72 ^b^**	**10.41 ^b^**
Prochloraz	**2.00 ^c^**	**3.06 ^c^**	**4.06 ^b^**	**6.72 ^b^**	**8.59 ^b^**
5% DMSO	3.15 ^b^	6.53 ^ab^	10.19 ^a^	15.0 ^a^	16.91 ^a^
Distilled water	6.00 ^a^	9.59 ^a^	12.44 ^a^	16.53 ^a^	18.22 ^a^

Note: D.A.I., days after inoculation. Numbers followed by the different letters showed the significant difference at 5% significance level by using Duncan’s test.

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
