# Peer review of "Use of Essential Oils for the Control of Anthracnose Disease Caused by Colletotrichum acutatum on Post-Harvest Mangoes of Cat Hoa Loc Variety"

_membranes, 2021, doi:10.3390/membranes11090719_

Round 1

Reviewer 1 Report

This manuscript reports an experiment on “Use of Essential Oils for the Control of Anthracnose Disease Caused by Colletotrichum acutatum on Post-Harvest Mangoes of Cat Hoa Loc Variety”. This is an interesting area of work. However, there is concern of the experiment mehodology which the clarity of methodology should be improved, especially the information of the part of the  plant has been extracted to produce EOs, the number of mangoe fruit for each treatment unit (ONE fruit ? ), temperature storage  and humidity of treated magoe fruit should be provided

The manuscript needs some corrections before the manuscript can be considered as an accepted manuscript.

Points for the authors to address:

  1. Section 2.5 ……. Peppermint and coriander fruit …. Is this correct as Peppermint and coriander are herbs ? Coriander fruits or seeds?
  2. Section 2.5 .. why didn’t all EOs were extracted using similar methods of extraction ? as Peppermint and coriander Eos were purchased from the company that possibly using different extraction method. Which part of the plant has been extracted ? Clarify this .. for section 2.5.
  3. Section 2.8 …… How many mangoes for each treatment unit? How many replications of this treatment? What was temperature storage after treatment?  Please provide additional information for section 2.8.
  4. Statistical analysis (Section 2.10) should be elaborated. One-way or Two-ways Anova ? What type of test to determine the significancy? At what level 0.01 or 0.05 ?
  5. Section 2.8  …..What was control solutions for these treatments ?. Please provide clear information for this section.
  6. Section 3.6 …..….. fungicide ….. What type of fungicide used as comparison for this study ? Clarify this information and provide the fungicide used in Materials and method section.

Author Response

  1. Section 2.5 ……. Peppermint and coriander fruit …. Is this correct as Peppermint and coriander are herbs ? Coriander fruits or seeds?

Response to reviewer: we can state that peppermint and coriander are herbs. Coriander essential oil used in this study was extracted from seeds. The term was corrected in manuscript.

  1. Section 2.5 .. why didn’t all EOs were extracted using similar methods of extraction ? as Peppermint and coriander Eos were purchased from the company that possibly using different extraction method. Which part of the plant has been extracted ? Clarify this .. for section 2.5.

Response to reviewer: we received information from the essential oil supplier that the method used for the extraction of peppermint and coriander seed essential oil was hydro-distillation, it was the same method we used to extract other essential oils in this study. The parts of the plants used for essential oil extraction were added in Section 2.5

  1. Section 2.8 …… How many mangoes for each treatment unit? How many replications of this treatment? What was temperature storage after treatment?  Please provide additional information for section 2.8.

Response to reviewer: there were three replicates for each treatment and one mango for each replicate. The mangoes were kept at 30°C after treatment. The information was added in the manuscript.

  1. Statistical analysis (Section 2.10) should be elaborated. One-way or Two-ways Anova ? What type of test to determine the significancy? At what level 0.01 or 0.05 ?

Response to reviewer: Data were processed by using the statistical software of SPSS 21.0 for one-way analysis of variance. The means of treatments were compared by using Duncan’s test at 5% level of significance. The information was added in Section 2.10.

  1. Section 2.8  …..What was control solutions for these treatments ?. Please provide clear information for this section.

Response to reviewer: there was no control treatment in this experiment, we observed mangoes before and after treatment. Observation of mango before treatment could be served as the control.

  1. Section 3.6 …..….. fungicide ….. What type of fungicide used as comparison for this study ? Clarify this information and provide the fungicide used in Materials and method section

Response to reviewer: fungicide used in this study was Prochloraz. The information was added in Section 3.6

Reviewer 2 Report

Please read your paper carefully, there are a lot of ortographic mistakes.

Author Response

Please read your paper carefully, there are a lot of ortographic mistakes

Response to reviewer: All ortographic mistakes were corrected according to the suggestion of the reviewer.